# Tsetse salivary glycoproteins are modified with paucimannosidic *N*-glycans, are recognised by C-type lectins and bind to trypanosomes

**Radoslaw P. Kozak[1‡], Karina Mondragon-Shem[2‡], Christopher Williams[3‡], Clair Rose[2], Samirah Perally[3], Guy Caljon[4], Jan Van Den Abbeele[5], Katherine Wongtrakul-Kish[1¤], Richard A. Gardner[1], Daniel Spencer[1], Michael J. Lehane[2], Álvaro Acosta-Serrano[2,3] ***

**1** Ludger Ltd., Culham Science Centre, Oxford, United Kingdom, **2** Department of Vector Biology, Liverpool School of Tropical Medicine, Liverpool, United Kingdom, **3** Department of Tropical Disease Biology, Liverpool School of Tropical Medicine, Liverpool, United Kingdom, **4** Laboratory of Microbiology, Parasitology and Hygiene (LMPH), University of Antwerp, Antwerp, Belgium, **5** Department of Biomedical Sciences, Institute of Tropical Medicine Antwerp, Antwerp, Belgium

¤ Current Address: Australian Research Council Centre of Excellence for Nanoscale Biophotonics, Macquarie University, Sydney, Australia

‡ These authors contributed equally to this work and share first authorship.

\* alvaro.acosta-serrano@lstmed.ac.uk

**Data Availability Statement:** All relevant data are within the manuscript and its Supporting Information files.

## Abstract

African sleeping sickness is caused by *Trypanosoma brucei*, a parasite transmitted by the bite of a tsetse fly. Trypanosome infection induces a severe transcriptional downregulation of tsetse genes encoding for salivary proteins, which reduces its anti-hemostatic and anti-clotting properties. To better understand trypanosome transmission and the possible role of glycans in insect bloodfeeding, we characterized the *N*-glycome of tsetse saliva glycoproteins. Tsetse salivary *N*-glycans were enzymatically released, tagged with either 2-amino-benzamide (2-AB) or procainamide, and analyzed by HILIC-UHPLC-FLR coupled online with positive-ion ESI-LC-MS/MS. We found that the *N*-glycan profiles of *T. brucei*-infected and naïve tsetse salivary glycoproteins are almost identical, consisting mainly (>50%) of highly processed $Man_3GlcNAc_2$ in addition to several other paucimannose, high mannose, and few hybrid-type *N*-glycans. In overlay assays, these sugars were differentially recognized by the mannose receptor and DC-SIGN C-type lectins. We also show that salivary glycoproteins bind strongly to the surface of transmissible metacyclic trypanosomes. We suggest that although the repertoire of tsetse salivary *N*-glycans does not change during a trypanosome infection, the interactions with mannosylated glycoproteins may influence parasite transmission into the vertebrate host.

**Funding:** This work was supported in part by the GlycoPar Marie Curie Initial Training Network 608295 (to KWK, DS and AA-S; www.ec.europa. eu), and a Ph.D. studentship from the Colombian Department of Science, Technology and Innovation (Colciencias) through the 'Francisco José de Caldas' scholarship programme (to KMS). The funders had no role in study design, data collection and analysis, decision to publish, or preparation of the manuscript.

**Competing interests:** The authors have declared that no competing interests exist.

## Author summary

In addition to helping the ingestion of a bloodmeal, the saliva of vector insects can modulate vertebrate immune responses. However, most research has focused on the salivary proteins, while the sugars (glycans) that modify them remain unexplored. Here we studied *N*-glycosylation, a common post-translational modification where sugar structures are attached to specific sites of a protein. Insect salivary *N*-glycans may affect how the saliva is recognized by the host, possibly playing a role during pathogen transmission. In this manuscript, we present the first detailed structural characterization of the salivary *N*-glycans in the tsetse fly *Glossina morsitans*, vector of African trypanosomiasis. We found that tsetse fly glycoproteins are mainly modified by simple *N*-glycans with short mannose modifications, which are recognised by mammalian C-type lectins (mannose receptor and DC-SIGN). Furthermore, we show that salivary glycoproteins bind to the surface of the trypanosomes that are transmitted to the vertebrate host; this opens up interesting questions as to the role of these glycoproteins in the successful establishment of infection by this parasite. Overall, our work represents a novel contribution towards the salivary *N*-glycome of an important insect vector, and towards the understanding of vector saliva and its complex effects in the vertebrate host.

## Introduction

Hematophagous insects have evolved special adaptations to ensure a successful bloodmeal from vertebrate hosts, key among which is their saliva. In these bloodfeeders, salivary proteins counteract the pain and itch of the bite, and fight host healing responses such as vasoconstriction and hemostasis [1]. At the same time, salivary compounds can elicit immune responses that are specific to each bloodfeeding species, which in turn can affect the pathogens they transmit [1]. Studies have also shown how vector salivary proteins are useful in disease control programs, either as markers of biting exposure [2] or as components of vaccines [3–5]. However, few studies have addressed the importance of the post-translational modifications in these proteins.

*N*-glycosylation is a highly common post- and co-translational modification where the carbohydrate chain is covalently attached to an asparagine residue on a protein containing the consensus sequon Asn-X-Thr/Ser [6]. A vast majority of secreted, non-cytosolic proteins are glycosylated [7]. *N*-glycans have a wide variety of functions, encompassing structural and modulatory properties, to the binding of other proteins and cell-cell interactions [8]. As they are secondary gene products, glycoprotein biochemistry varies not only between species but also between cell types in the same organism. *N*-glycosylation can affect protein folding, protein stability, ligand binding, and protein antigenicity [6, 9].

In the discipline of glycobiology, insects remain a neglected area of study. Most research has focused on the model fly *Drosophila melanogaster*, where *N*-glycans in particular are important for aspects such as cell adhesion, morphogenesis, and locomotion [10]. As the field of invertebrate glycobiology expands, more work is published on organisms like butterflies, bees and some dipterans [11, 12]; these confirm that insect cells mainly produce oligo- and paucimannosidic *N*-glycans, with some hybrid- and complex-type structures in some species [13]. Compared to *Drosophila*, there has been little research into the *N*-glycans of blood feeding insects, where the salivary glycoproteins of disease vectors are of special interest considering how important saliva is during hematophagy and vector-host-pathogen interactions.

Tsetse flies are medically and economically important disease vectors in sub-Saharan Africa, where they transmit the parasites that cause human and animal trypanosomiases. Both male and female tsetse flies are obligate blood feeders. Upon feeding from an infected mammalian host, they ingest bloodstream trypomastigotes, which transform into procyclic trypomastigotes in the fly's midgut [14]. They subsequently multiply as epimastigotes in the salivary gland, and then transform into metacyclic trypomastigotes, which are co-transmitted with saliva during the next blood meal and can infect a new vertebrate host [15]. Uninfected tsetse flies can inoculate ~4 μg of salivary proteins [16], which can alter host immune responses [17]. However, trypanosome infection induces a severe (~70%) transcriptional downregulation of the tsetse genes encoding for salivary proteins, affecting their anti-hemostatic and anti-clotting properties [18]. In this work we characterized the salivary *N*-glycome of the tsetse fly *Glossina* spp., comparing naïve and trypanosome-infected flies. Using highly sensitive liquid chromatography and mass spectrometry, we revealed the presence of several salivary glycoproteins in tsetse saliva, with oligosaccharides composed mainly of pauci-mannose and high-mannose *N*-glycans. Our work presents the first structural analysis of salivary glycans from tsetse flies.

## Methods

### Ethics statement

Experiments using laboratory rodents were carried out in strict accordance with all mandatory guidelines (European Union directives, including the Revised Directive 2010/63/EU on the Protection of Animals Used for Scientific Purposes and the Declaration of Helsinki) and were approved by the ethical committee of the Institute of Tropical Medicine Antwerp, Belgium (PAR011-MC-M-Tryp).

### Tsetse fly saliva

*Glossina morsitans morsitans* adults were obtained from the tsetse insectary at the Liverpool School of Tropical Medicine (UK). Flies were maintained at 26°C and 65–75% relative humidity, and fed for 10 min every two days on sterile, defibrinated horse blood (TCS Biosciences Ltd., Buckingham, UK). Tsetse flies were chilled at 4°C for 20 min, after which the salivary glands were dissected on ice and placed in sterile PBS. The salivary glands were centrifuged at 4°C, 10,000 rpm for 10 min, and the supernatant collected and stored at -20°C.

### Tsetse infection

Teneral male *G. m. morsitans* were infected with *Trypanosoma brucei brucei* by combining an aliquot of *T. brucei* (TSW196 strain [19]) infected rat blood with sterile defibrinated horse blood. Flies were bloodfed every two days for 4 weeks until infection of salivary glands was achieved. Saliva was extracted using sterile PBS and stored at -20°C. For evaluation of salivary protein binding to metacyclic trypanosomes, flies were infected with *T. brucei* AnTAR1 [16] in defibrinated horse blood supplemented with 10 mM reduced L-glutathione.

### Enzymatic deglycosylation for glycoprotein detection

Tsetse salivary proteins were treated with peptide-N-glycosidase F (PNGase F, New England Biolabs), which cleaves all *N*-linked glycans except those with an α-1,3 fucose modification of the chitobiose core [20]. Deglycosylation was done according to the manufacturer's instructions: briefly, 1x glycoprotein denaturing buffer (5% SDS, 0.4 M DTT) was added to 10 μg of *G. m. morsitans* salivary proteins and incubated at 100°C for 10 min. 1x G7 reaction buffer

(0.5 M sodium phosphate pH 7.5), 1% NP40 and 1 μl PNGase F were added and incubated at 37°C overnight.

Salivary samples were additionally treated with Endoglycosidase H (Endo H; New England Biolabs), which cleaves the chitobiose core of high-mannose and some hybrid *N*-linked glycans [21]. Deglycosylation conditions were the same as described for PNGase F treatment, with an addition of NP40 and the G5 Reaction buffer (50 mM sodium citrate, pH 5.5) used instead of G7; 5000 units of Endo H were used per reaction. Egg albumin (Sigma) was treated in parallel as digestion control for both enzymes.

## SDS-PAGE analysis of tsetse salivary proteins

Salivary proteins were resolved on 12% polyacrylamide gels. InstantBlue (Expedeon) and Pierce Glycoprotein Staining kit (ThermoFisher) were used for protein and glycan staining, respectively.

## Western blotting

~2 μg of *G. m. morsitans* salivary proteins were treated with PNGase F (New England Biolabs), resolved by SDS-PAGE and transferred onto a PVDF membrane (GE Healthcare) at 90V for 1 hour. After verification of transfer with Ponceau Red (Sigma-Aldrich), the membrane was blocked for 1 hour (PBS-T [Sigma, US] containing 5% skim milk powder [Sigma]) and incubated in 1:10,000 rabbit-anti-*G. m. morsitans* saliva antibody overnight at 4°C. Membranes were washed and probed at room temperature for 1 hour with 1:20,000 HRP-labelled goat-anti-rabbit antibody (ThermoFisher). Super Signal West Dura substrate (ThermoFisher) was used for detection.

## Concanavalin A (Con A) blotting

~1 μg of tsetse saliva was treated with PNGase F (New England Biolabs) as described above. After digestion, proteins were fractionated on a 12.5% polyacrylamide gel, transferred onto a PVDF membrane, and placed in blocking buffer (1% BSA-PBS-Tw 20 [Sigma]) at 4°C overnight. Following this, the membrane was incubated with 1 μg/ml biotinylated Con A lectin (Vector Labs, Peterborough) in blocking buffer at room temperature for 1 hour. After washing, the membrane was incubated with 1:100,000 streptavidin-HRP (Vector Labs). SuperSignal West Pico Chemiluminescent substrate (ThermoFisher) was used for detection. Egg albumin (1 μg) was used as a positive control for enzymatic deglycosylation and Con A detection.

## Overlay assays with C-type lectins

Saliva samples were deglycosylated overnight with PNGase F (New England Biolabs), and the processed for SDS-PAGE and transfer to PVDF membrane as described above. After blocking overnight with 1% BSA (Sigma), the membranes were incubated with CTLD4-7Fc (0.5 μg/μl) or DC-SIGN (0.5 μg/μl) (R&D Systems) for 1 hour, washed, and then incubated with anti-human IgG conjugated to HRP for 1 hour. After washing, WestDura substrate (ThermoFisher Scientific) was used for chemiluminescent detection.

## Evaluation of salivary protein binding to metacyclic trypanosomes

Metacyclic trypanosomes were isolated from the outflow of salivary glands from infected tsetse. Parasites were subjected to two subsequent washes with 1.2 mL PBS. Aliquots of those washes were kept for polyacrylamide gel electrophoresis and Western blot analysis. A final pellet of $10^5$ parasites was obtained, resuspended in 50 μL PBS and stored at -80°C. As a control,

aliquots of $10^5$ bloodstream form parasites were prepared following isolation from the blood of infected mice by the mini anion exchange centrifugation technique (mAECT). Parasite washes and lysates were separated on NuPAGE 12% Bis-Tris Mini Protein Gels (Invitrogen) at 125 V in an XCell Surelock Mini-Cell (Invitrogen). Gels were used for silverstaining (PageSilver kit, Fermentas) or transferred to Hybond-C nitrocellulose membranes (GE Healthcare). After overnight blocking in PBS with 5% non-fat dry milk and 0.1% Tween 20, blots were incubated overnight at 4˚C with 1 μg/mL purified rabbit polyclonal IgGs (an admix of anti-G. *morsitans* saliva IgG and IgGs against the purified recombinant proteins Tsal1, Tsal2, Tag5, 5'Nucleotidase-related protein and Gmmsgp3). As a negative control, blots were incubated with 1 μg/mL purified IgGs from preimmune rabbits. After washing with the blocking solution, membranes were incubated for 1 hour at room temperature with 1:10,000 diluted peroxidase-conjugated goat anti-rabbit IgG (Sigma, A6154). After 4 washes in PBS with 0.1% Tween 20, blots were developed with Immobilon chemiluminescent horseradish peroxidase substrate (Millipore) and exposed to an autoradiography film (GE Healthcare).

## Mass spectrometry analysis of salivary proteins

To identify the glycoproteins that were susceptible to PNGase F cleavage, 10 μg of salivary proteins were resolved in a 12% polyacrylamide gel and Coomassie stained. Bands of interest were sent to the Dundee University Fingerprints Proteomics Facility, where they were subjected to in-gel trypsination and then alkylated with iodoacetamide. Peptides were analyzed by liquid chromatography-tandem mass spectrometry (LC-MS/MS) in a Thermo LTQ XL Linear Trap with a nano-LC.

The data was supplied in MASCOT format. The gi numbers for the top hits in each band were searched in NCBI Protein (http://www.ncbi.nlm.nih.gov/protein) to yield the FASTA format of the protein sequence. This was then queried in PROWL (http://prowl.rockefeller.edu/) to reveal the predicted molecular weight and also to predict tryptic peptides in the sequence. The FASTA protein sequence was also queried in the SignalP 4.0 Server software[187] to predict the signal peptide location and NetNGlyc 1.0 (http://www.cbs.dtu.dk/services/NetNGlyc/) to reveal potential *N*-glycosylation sites.

## Release of tsetse salivary *N*-linked glycans for structural analysis

*N*-linked glycans were released by in-gel deglycosylation using PNGase F (QA Bio) as described in [22]. In addition, tsetse saliva was treated with peptide-N-glycosidase A (PNGase A, New England Biolabs), which releases all *N*-linked glycans, including those with an α-1,3 fucose modification of the chitobiose core [20]. Briefly, peptides were released from gel pieces by overnight incubation at 37˚C with trypsin in 25 mM ammonium bicarbonate. The supernatant was dried, re-suspended in water and heated at 100˚C for 10 min to deactivate the trypsin. After drying by vacuum centrifugation, the tryptic peptide mixture was incubated with PNGase A in 100 mM citrate/phosphate buffer (pH 5) for 16 hours at 37˚C [23]. PNGase F and PNGase A released *N*-glycans were separated from protein and salts by LudgerClean Protein Binding Plate (Ludger Ltd., Oxfordshire, UK). Glycans pass straight through the protein binding membrane. Wells are flushed with extra water to ensure full recovery, and then dried by vacuum centrifugation prior to fluorescent labelling.

## Release of tsetse salivary *O*-linked glycans by hydrazinolysis

Samples from tsetse saliva and fetuin (as positive control) were incubated in anhydrous hydrazine (Ludger) at 60˚C for 6 h as described in [24]. Samples were then chilled, re-*N*-acetylated

in 0.1 M sodium bicarbonate solution and acetic anhydride, and labelled with 2-aminobenzamide (2AB) as indicated below.

## Fluorescent labelling (2-aminobenzamide or procainamide) and purification

Released *N*-glycans were fluorescently labelled via reductive amination by reaction with 2-aminobenzamide (2-AB) using a Ludger 2-AB Glycan Labelling Kit (Ludger Ltd., Oxfordshire, UK), or a Ludger Procainamide Glycan Labelling Kit (Ludger Ltd., Oxfordshire, UK), both containing 2-picoline borane. Released glycans were incubated with labelling reagents for 1 hour at 65˚C [25]. 2-AB labelled glycans were cleaned up using LudgerClean T1 cartridges (Ludger Ltd., Oxfordshire, UK). Procainamide labelled glycans were cleaned up using LudgerClean S cartridges (Ludger Ltd., Oxfordshire, UK). In both cases, fluorescently labelled glycans were eluted with 1 mL water. Samples were evaporated and then re-suspended in water for further analysis.

## Exoglycosidase sequencing

Exoglycosidase digestion was performed according to [22]. The released, 2-AB labelled *N*-glycans were incubated with exoglycosidases at standard concentrations in a final volume of 10 μL in 50 mM sodium acetate (except for incubations with JBM, where 250 mM sodium phosphate, pH 5.0 was used) for 16 hours at 37˚C. Glycans were incubated with different exoglycosidases in different sequences: (i) *Streptococcus pneumoniae* β-N-acetylglucosaminidase (GUH, New England Biolabs); (ii) Jack bean α-(1–2,3,6)-Mannosidase (JBM, QA-Bio); (iii) Bovine kidney α-(1–2,3,4,6)-Fucosidase (bkF; Sigma-Aldrich) and Jack bean α-(1–2,3,6)-Mannosidase (JBM, QA-Bio). The bkf enzyme contains approximately ≤0.2% β-*N*-acetylglucosaminidase and ≤0.1% of α-mannosidase and β-galactosidase, thus combination of bkf and JBM results in necessary to achieve full and final sequencing. After digestion, samples were separated from the exoglycosidases by binding onto a LudgerClean Post-Exoglycosidase clean-up plate (Ludger Ltd., Oxfordshire, UK) for 60 min followed by elution of the glycans from the plate with water.

## HILIC-UHPLC-FLR analysis

2-AB labelled samples were analyzed by HILIC-UPLC using an ACQUITY UPLC BEH-Glycan column (1.7 μm, 2.1 x 150 mm) at 60˚C on a Dionex UltiMate 3000 UHPLC instrument (Thermo, UK) with a fluorescence detector (λex = 250 nm, λem = 428 nm), controlled by Chromeleon data software version 6.8. Gradient conditions were: 0 to 53.5 min, 24% A (0.4 mL/min); 53.5 to 55.5 min, 24 to 49% A (0.4 mL/min); 55.5 to 57.5 min, 49 to 60% A (0.4 to 0.25 mL/min); 57.5 to 59.5 min, 60% A (0.25 mL/min); 59.5 to 65.5 min, 60 to 24% A (0.4 mL/min); 65.5 to 66.5 min, 24% A (0.25 to 0.4 mL/min); 66.5 to 70 min 24% A (0.4 mL/min). Solvent A was 50 mM ammonium formate; solvent B was acetonitrile (Acetonitrile 190 far UV/gradient quality; Romil #H049, Charlton Scientific). Samples were injected in 24% aqueous/76% acetonitrile with an injection volume of 25 μL. Chromeleon software retention index function with a cubic spline fit was used to allocate glucose units (GU) values to peaks [26]. 2-AB labelled glucose homopolymer (Ludger Ltd., Oxfordshire, UK) was used as a system suitability standard as well as an external calibration standard for GU allocation on the system [22, 26].

## Online Hydrophilic Interaction Liquid Chromatography-Solid Phase Extraction LC-MS analysis

Procainamide labelled glycans were prepared in 0.1% TFA (v/v) in 78% acetonitrile (v/v) and desalted on-line using hydrophilic interaction liquid chromatography solid phase extraction

prior to direct elution into the mass spectrometer. Samples were applied onto a HILIC trapping column (MilliporeSigma 1.50484.0001 SeQuant HPLC Guard Column with ZIC-HILIC [5 μm] Sorbent Packing Media, 5 x 0.3mm) at a flow rate of 1.5 μl/min using an UltiMate 3000 LC (Thermo Scientific, Massachusetts). The trap was washed for 4 min with 0.1% formic acid in 90% ACN (v/v) followed by elution of procainamide labelled glycans using an isocratic gradient of 0.1% formic acid in 27% ACN (v/v) for 12 min. Glycans were analysed by electrospray ionization tandem mass spectrometry (ESI-MS) on an amaZon speed ETD ion trap MS (Bruker, Massachusetts) in positive ion mode with a nitrogen flow of 10 L/min and capillary voltage of 4500 V.

### LC-ESI-MS and LC-ESI-MS/MS analysis

Procainamide labelled samples were analyzed by HILIC-UHPLC-ESI-MS with fluorescence detection ($\lambda_{ex}$ = 310 nm, $\lambda_{em}$ = 370 nm), using the same UHPLC conditions as detailed above for 2-AB analysis, with the exception of column temperature (set to 40°C). ESI-MS and MS/MS detection were carried out using an amaZon speed ETD ion trap MS (Bruker, Massachusetts) as above, with the top three precursors ions selected for MS/MS fragmentation.

## Results

### Identification of tsetse salivary glycoproteins

Bioinformatic analysis was performed to identify potential glycosylation sites on tsetse salivary proteins, looking at proteins having the Asn-X-Ser/Thr sequons. The NetNGlyc server prediction tool [27] identified that 72% of *Glossina* proteins have at least one potential *N*-glycosylation site (S1 Table). However, although the consensus sequence is a prerequisite for the addition of *N*-glycans to the asparagine, it does not guarantee their glycosylation status *in vivo*. The presence of glycoproteins was then confirmed by Schiff's staining of SDS-PAGE separated proteins, which indicated several glycoproteins migrating with different apparent molecular masses (S1 Fig). Tsetse salivary proteins contained in the migrating bands were identified by mass spectrometry (S2 and S3 Tables) and complemented the assignment using published data [17, 18, 28–30].

To identify the type of glycosylation (*N*- or *O*-linked) present in tsetse salivary glycoproteins, we first released the *N*-glycans and analyzed the deglycosylated proteins by SDS-PAGE fractionation and mass spectrometry. Treatment with PNGase F revealed several proteins that showed an electrophoretic shift after deglycosylation (Fig 1). Furthermore, PNGase F treatment allowed the fractionation and visualization of salivary proteins with an apparent high molecular mass (Fig 1, lane 2). Proteomic analysis revealed 4 bands with notable shift in electrophoretic migration: 5' Nucleotidase-related protein (5'Nuc), TSGF 2/Adenosine deaminase, TSGF 1, and Tsal 1/2. No proteins were detected by Schiff's staining (S1 Fig) after treatment, suggesting that the main type of sugars linked to tsetse salivary glycoproteins are likely to be *N*-glycans.

To better characterize the type of *N*-glycosylation present, salivary glycoproteins were treated with Endo H (S2 Fig), which cleaves *N*-glycans with at least 3 mannose residues where the α1,6 mannose branch is attached to another mannose [31]. These results showed an SDS-PAGE band migration similar to that observed following PNGase F treatment (Fig 1). Mass spectrometric protein identification of the main migrating bands also detected similar glycoproteins, and the protein identifications, number of potential glycosylation sites, and peptide coverage is shown in S3 Table. These comprise the most abundant glycoproteins in the saliva of the tsetse fly; however, we cannot discount the existence of proteins in very low abundance that our methods could not detect.

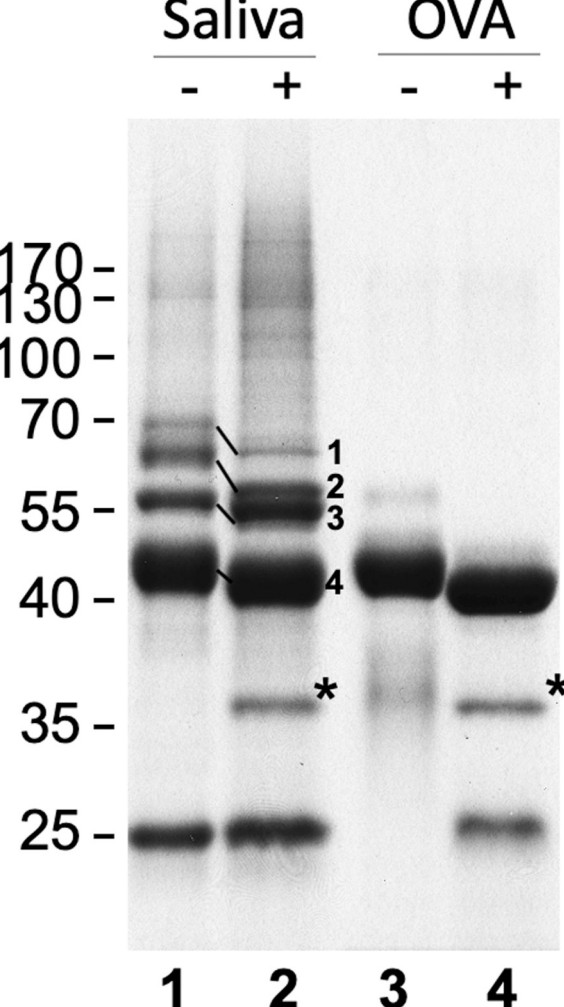

**Fig 1. Analysis of _G. morsitans_ salivary glycoproteins.** 10 μg of _G. morsitans_ salivary proteins (lanes 1 and 2) and 10 μg of egg albumin (lanes 3 and 4) were incubated overnight with (2 and 4) and without (1 and 3) PNGase F. After digestion, proteins were resolved by SDS-PAGE and Coomassie blue-stained. There was a notable shift in migration in 4 bands following PNGase F treatment. After in-gel trypsinization and MALDI-TOF MS analysis these bands were identified as 5' Nucleotidase (1), TSGF 2/Adenosine deaminase (2), TSGF 1 (3), and Tsal 1/2 (4). *, PNGase F enzyme.

## Structural characterization of _G. morsitans_ salivary _N_-glycans

For full structural characterization of the _N_-glycome of _G. m. morsitans_ saliva, _N_-linked glycans where released by PNGase F digestion, purified and tagged with 2-AB, a fluorescent label for chromatographic detection. HILIC-UHPLC-FLR analysis revealed 13 peaks that correspond to a variety of potential high mannose and hybrid _N_-glycan structures (Fig 2A). The peak of highest intensity (abundance) corresponds to the core structure $Man_3GlcNAc_2$-2AB. After treatment with PNGase A, which cleaves all types of _N_-glycans (including those with core fucose residues in an α-1,3 linkage to the reducing end GlcNAc), the profile of oligosaccharides did not show any difference to the one obtained by PNGase F digestion (S3 Fig). This suggests the absence of α-1,3 core fucosylated structures in _G. m. morsitans_ saliva, and indicates that the single core fucosylated structure identified (peak 3, Fig 2A) probably carries an α-1,6 linked core fucose.

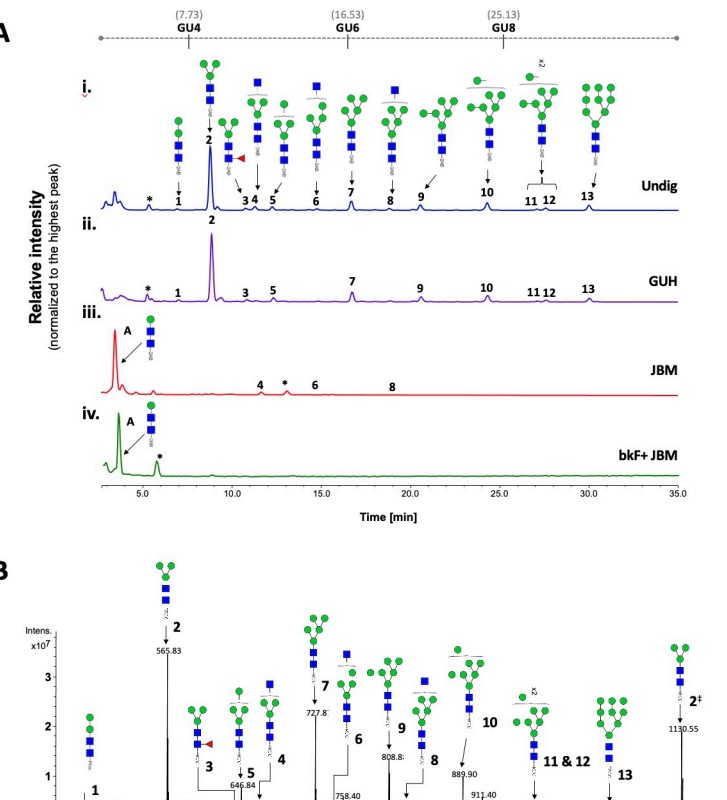

**Fig 2. Tsetse fly salivary glycoproteins are composed mainly of paucimannose and oligomannose N-glycans.** (A) Profile of salivary *N*-glycans from teneral (young, unfed) flies, before and after digestion with exoglycosidases. Aliquots of the total PNGase F-released 2-AB-labeled *N*-glycan pool were either undigested (i) or incubated with a range of exoglycosidases (ii-iv). (i) Undig, before digestion; (ii) GUH, *Streptococcus pneumoniae* in *E. coli* β-N-acetylglucosaminidase; (iii) JBM, Jack bean α-Mannosidase; (iv) bkF, Bovine kidney α-fucosidase. Following digestion, the products were analyzed by HILIC-UHPLC. Peaks labelled A correspond to the product of complete digestion with JBM; those labelled with an asterisk refer to buffer contaminants. The percent areas and structures of the different *N*-glycans are listed in Table 1. (B) Positive-ion ESI-MS spectrum of procainamide-labelled *N*-glycans from teneral tsetse fly saliva. Numbers refer to the structures in Table 1. The dagger symbol (‡) refers to *m/z* 1130.55 as [M+2H]$^{2+}$ ion; the appearance of the Man₃GlcNAc₂-Proc as singly and doubly charged ion in positive mode, reflects on its high relative abundancy (~54%) in this sample. Green circle, mannose; blue square, *N*-Acetylglucosamine; red triangle, fucose; Proc, procainamide. GU, glucose homopolymer ladder. [22].

To further characterize the structure of these *N*-linked glycans, samples were treated with exoglycosidases of different specificities to confirm monosaccharide linkages: β-*N*-Acetylglucosaminidase (GUH; Fig 2A–ii), Jack bean α-(1–2,3,6)-mannosidase (JBM; Fig 2A–iii), bovine kidney fucosidase (bkF; Fig 2A–iv). GUH resulted in a reduction of peaks 4, 6 and 8, indicating the presence of a terminal non-reducing β-GlcNAc in these structures. This was further corroborated after JBM digestion, where most glycans (with the exception of peaks 4, 6 and 8, see Table 1) were completely digested as shown by the appearance of a new peak identified as βMan-βGlcNAc-βGlcNAc-2AB (Fig 2A–iii). Finally, sequential treatment with bkF followed by JBM resulted in the loss of all peaks in the chromatogram. Peak information after enzymatic treatment is further detailed in Table 1.

For secondary confirmation of the salivary *N*-glycan structures by mass spectrometry, released oligosaccharides were labelled with procainamide and then analyzed by positive-ion ESI-MS and ESI-MS/MS. The resulting mass spectra confirms the findings by HILIC-UHPLC

**Table 1. Exoglycosidase digestion of 2-AB-labelled glycans released from teneral fly saliva by PNGase F.**

| HILIC-UPLC | GU (2-AB) | Structure | Enzymes used[a] | | | |
|---|---|---|---|---|---|---|
| Peak ID | | | Undig | GUH | JBM | bkF + JBM |
| A | 2.55 | Man$_1$GlcNAc$_2$[b] | 0.00 | 0.00 | 90.64 | 91.44 |
| 1 | 3.84 | Man$_2$GlcNAc$_2$ | 1.59 | 2.85 | 0.00 | 0.00 |
| 2 | 4.34 | Man$_3$GlcNAc$_2$ | 55.35 | 52.06 | 0.00 | 0.00 |
| 3 | 4.83 | Man$_2$Fuc1GlcNAc$_2$ | 2.23 | 4.05 | 0.00 | 0.00 |
| 4 | 4.96 | GlcNAc$_1$Man$_3$GlcNAc$_2$ | 4.06 | 0.00 | 3.32 | 0.00[c] |
| 5 | 5.18 | Man$_4$GlcNAc$_2$ | 3.09 | 5.55 | 0.00 | 0.00 |
| 6 | 5.74 | GlcNAc$_1$Man$_4$GlcNAc$_2$ | 1.53 | 0.00 | 0.46 | 0.00 |
| 7 | 6.17 | Man$_5$GlcNAc$_2$ | 8.41 | 10.04 | 0.00 | 0.00 |
| 8 | 6.66 | GlcNAc$_1$Man5GlcNAc$_2$ | 1.76 | 0.00 | 0.15 | 0.00 |
| 9 | 7.09 | Man$_6$GlcNAc$_2$ | 5.94 | 6.38 | 0.00 | 0.00 |
| 10 | 7.97 | Man$_7$GlcNAc$_2$ | 7.70 | 8.12 | 0.00 | 0.00 |
| 11 | 8.71 | Man$_8$GlcNAc$_2$ | 1.37 | 2.87 | 0.00 | 0.00 |
| 12 | 8.84 | Man$_8$GlcNAc$_2$ | 2.13 | 3.08 | 0.00 | 0.00 |
| 13 | 9.54 | Man$_9$GlcNAc$_2$ | 4.86 | 5.01 | 0.00 | 0.00 |

[a]Numbers are percentage areas;

[b]Digestion product only.

[c]Complete digestion after incubation with bkF + JBM is likely due to the presence of contaminating (~0.2%) β-*N*-acetylglucosaminidase activity in commercial bfk enzyme. Corresponding glycan structures are illustrated in Fig 2. Undig, whole glycan pool before digestion; GUH, *Streptococcus pneumonia* in *E coli* β-N-acetylglucosaminidase; JBM, Jack bean α-mannosidase; bkF, Bovine kidney α-fucosidase. GU, glucose homopolymer ladder [22].

analysis, showing the presence of 12 [M+H]$^{2+}$ ions with *m/z* 565.74, 646.74, 727.79, 808.81, 889.84, 970.87, 1051.90 (corresponding to Man$_{3-9}$GlcNAc$_2$-Proc), in addition to three complex type glycans with truncated antenna; i.e. Man$_3$GlcNAc$_2$Fuc-Proc, Man$_3$GlcNAc$_3$-Proc, and Man$_4$GlcNAc$_3$-Proc (Fig 2B and S4 Table). Structural topology was confirmed by positive ion ESI-MS/MS fragmentation spectra, including the most abundant species Man$_3$GlcNAc$_2$-Proc ([*m/z*]$^+$ 1130.49) as well as Man$_3$GlcNAc$_2$Fuc-Proc ([*m/z*]$^+$ 1276.57) and Man$_3$GlcNAc$_3$-Proc ([*m/z*]$^+$ 1333.59) (S4 Fig and S5 Table). Details of all *N*-glycans and the MS/MS diagnostic ions used in their identification can be found in S4 Table. Overall, these results suggest that tsetse salivary *N*-linked glycans consist mainly of the highly processed Man$_3$GlcNAc$_2$ in addition to several other paucimannose, oligomannose, and few hybrid-type *N*-glycans.

## *N*-glycosylation profile of tsetse saliva remains unaffected during a trypanosome infection

Since *T. brucei* infection affects the composition of tsetse saliva [18], we investigated if infection changes salivary *N*-glycosylation as well. Initially, we compared the salivary profiles of teneral (newly emerged adults) and naïve (uninfected) flies with those that had either a salivary gland or a midgut infection with *T. brucei* by SDS-PAGE (S5 Fig). When samples were normalized by protein concentration, there were no major changes in the profile of salivary proteins in the different physiological states.

To determine whether *T. b. brucei* infection alters the structure of salivary *N*-glycans, oligosaccharides were released with PNGase F, labelled with procainamide, and analyzed by HILIC coupled with ESI-MS. Fig 3A shows the HILIC chromatogram, where no change is observed in the saliva *N*-glycan population following infection of the salivary gland. A comparison of the relative percentage areas of glycan peaks identified in teneral, naïve and infected fly saliva

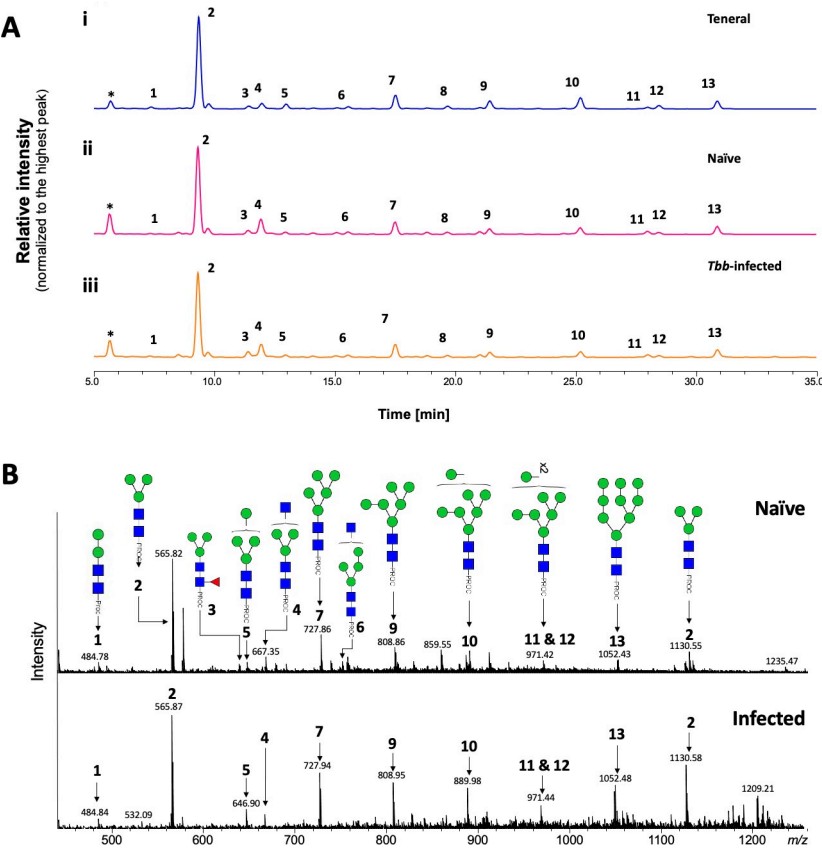

**Fig 3. Analysis of tsetse salivary *N*-linked glycans in teneral, naïve and trypanosome-infected flies.** (A) Comparison of HILIC-UHPLC profiles of salivary *N*-glycans released by PNGase F. Analysis of 2AB-labelled glycans from (i) teneral, (ii) naïve, and (iii) trypanosome-infected saliva. Relative abundances are indicated in Table 2. Tbb, *Trypanosoma brucei brucei*. (B) Positive-ion ESI-MS analysis of procainamide labelled *N*-glycans from adult naïve and trypanosome-infected saliva. Spectra are shown for naïve (top) and trypanosome-infected (bottom) saliva. Numbers refer to the structures shown in Table 1. Green circle, mannose; blue square, *N*-Acetylglucosamine; red triangle, fucose; Proc, procainamide. Peaks labelled with an asterisk refer to buffer contaminants.

(Table 2) showed a slight variation in the abundance of some structures (Table 2: peaks 4, 9, 10, 13), potentially an effect of blood ingestion.

## *Trypanosoma* infection does not alter immune reactivity of tsetse salivary glycoproteins with antibodies

Next, we investigated whether a trypanosome infection alters the immune reactivity of tsetse salivary glycoproteins. By immunoblotting, we compared the saliva of flies with midgut or salivary gland infection, before and after treatment with PNGase F (Fig 4A and 4B). After probing with a polyclonal anti-*G. morsitans* saliva rabbit serum, recognition of control *G. morsitans* saliva before and after cleavage of the glycans appears unaffected. However, during salivary gland infection the polyclonal serum only detected the high molecular weight proteins (100 kDa-130 kDa) after glycans were cleaved. The effect is more readily seen here possibly due to the downregulation of other salivary proteins during infection, and seems to be concealed both in the saliva of naïve flies and those with midgut infection. Interestingly, saliva from trypanosome-infected flies displayed an antigenic ~20 kDa band that is faintly seen by SDS-PAGE (S5 Fig), and is absent from uninfected saliva following Western blotting. We

**Table 2. Comparison of relative abundance of *N*-glycans released by PNGase F from Teneral, Naïve and Infected Fly Saliva.** GU, glucose units. Data correspond to 2-AB labelled samples.

| HILIC-UPLC Peak ID | GU (2-AB) | Structure | Enzymes used[a] | | | |
|---|---|---|---|---|---|---|
| | | | Undig | GUH | JBM | bkF + JBM |
| A | 2.55 | $Man_1GlcNAc_2$[b] | 0.00 | 0.00 | 90.64 | 91.44 |
| 1 | 3.84 | $Man_2GlcNAc_2$ | 1.59 | 2.85 | 0.00 | 0.00 |
| 2 | 4.34 | $Man_3GlcNAc_2$ | 55.35 | 52.06 | 0.00 | 0.00 |
| 3 | 4.83 | $Man_2Fuc_1GlcNAc_2$ | 2.23 | 4.05 | 0.00 | 0.00 |
| 4 | 4.96 | $GlcNAc_1Man_3GlcNAc_2$ | 4.06 | 0.00 | 3.32 | 0.00[c] |
| 5 | 5.18 | $Man_4GlcNAc_2$ | 3.09 | 5.55 | 0.00 | 0.00 |
| 6 | 5.74 | $GlcNAc_1Man_4GlcNAc_2$ | 1.53 | 0.00 | 0.46 | 0.00 |
| 7 | 6.17 | $Man_5GlcNAc_2$ | 8.41 | 10.04 | 0.00 | 0.00 |
| 8 | 6.66 | $GlcNAc_1Man5GlcNAc_2$ | 1.76 | 0.00 | 0.15 | 0.00 |
| 9 | 7.09 | $Man_6GlcNAc_2$ | 5.94 | 6.38 | 0.00 | 0.00 |
| 10 | 7.97 | $Man_7GlcNAc_2$ | 7.70 | 8.12 | 0.00 | 0.00 |
| 11 | 8.71 | $Man_8GlcNAc_2$ | 1.37 | 2.87 | 0.00 | 0.00 |
| 12 | 8.84 | $Man_8GlcNAc_2$ | 2.13 | 3.08 | 0.00 | 0.00 |
| 13 | 9.54 | $Man_9GlcNAc_2$ | 4.86 | 5.01 | 0.00 | 0.00 |

suggest these probably represent proteolytic products of salivary proteins that are formed as a result of the trypanosome infection in the gland [32]. Fig 4C shows that Con A equally recognised salivary glycoproteins from either naïve or trypanosome-infected flies.

## Glycosylated salivary proteins bind onto metacyclic trypanosomes

Binding of saliva proteins to metacyclic trypanosomes was evaluated by isolating and washing metacyclics from *T. brucei*-infected *G. morsitans* salivary glands. Presence of saliva proteins in the washes and on metacyclic trypanosomes was revealed by Western blot using rabbit anti-*G. morsitans* saliva IgGs (Fig 5). Specificity of salivary protein detection was assured by including rabbit pre-immune IgGs and a lysate of an equal number of bloodstream trypanosomes as

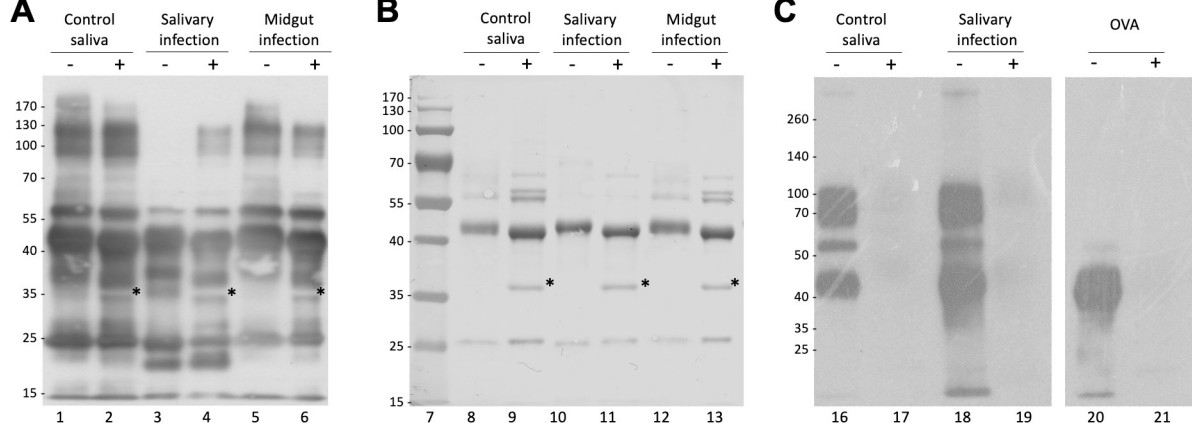

**Fig 4. Analysis of the effects of infection on immunogenicity of tsetse fly saliva.** (A) 2 μg of *G. m. morsitans* salivary proteins were treated (+) or untreated (-) with PNGase F, fractionated by SDS-PAGE, transferred onto a PVDF membrane, and probed with an anti-*G. m. morsitans* saliva antibody. (B) Uniform protein loading for Western blot was confirmed by nigrosine staining of proteins transferred to PVDF membrane. (C) Con A blotting analysis of tsetse salivary glycoproteins from naïve and trypanosome-infected flies. OVA, egg albumin positive control. Asterisk indicates PNGase F enzyme band.

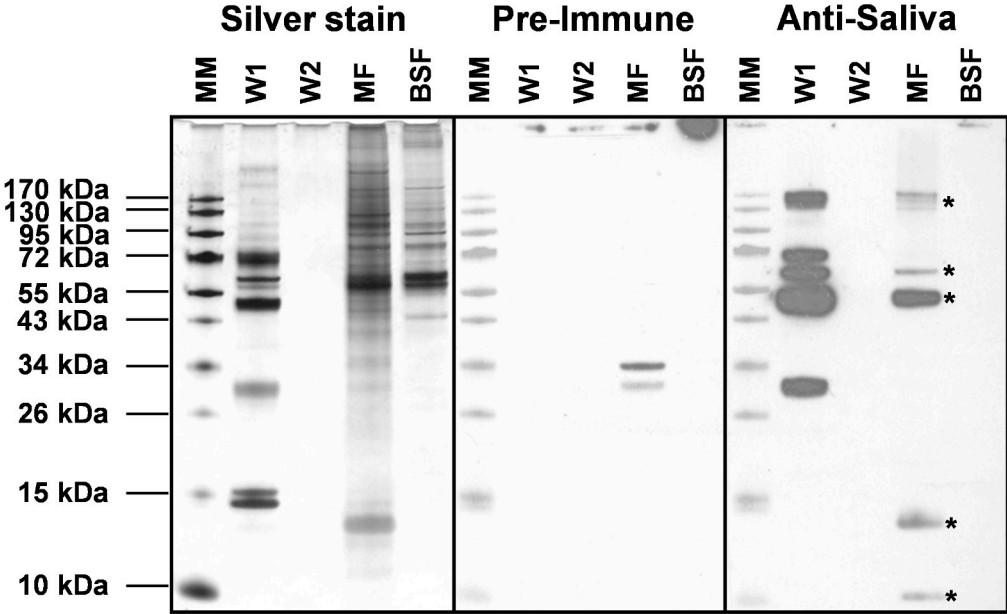

**Fig 5. Analysis of the binding of salivary proteins to metacyclic trypanosomes.** Silver stained protein profiles and Western blot analysis to detect the presence of salivary proteins on tsetse salivary gland-derived trypanosomes. Two subsequent washes (W1 and W2) of metacyclic parasites (MF, equivalent of $3 \times 10^4$ parasites loaded on gel) isolated from infected tsetse fly salivary glands and a corresponding sample of trypanosomes purified from mouse blood (BSF, equivalent of $3 \times 10^4$ parasites loaded). Protein bands were revealed with purified rabbit anti-*G. morsitans* saliva IgGs and pre-immune IgGs as a control, and development with peroxidase-coupled goat anti-rabbit IgG. Asterisk indicates salivary proteins in the metacyclic trypanosome lysate.

controls. The second parasite wash was devoid of detectable levels of saliva proteins, whereas the metacyclic parasite lysates contained various protein bands that were specifically recognized by the immune serum (Fig 5). Based on their apparent molecular masses on SDS-PAGE and mass spectrometry identification (Figs 1 and S1), these components likely correspond to 5'Nucleotidase-related protein Gmmsgp3, TSGF and the Tsal glycoproteins. Two additional unidentified < 15 kDa bands were specifically detected using the anti-saliva IgGs. Interestingly, the abundant, non-glycosylated TAg5 protein in saliva was found not to bind to the metacyclic trypanosome surface.

## *N*-glycans from *G. morsitans* salivary glycoproteins are recognised by the Mannose Receptor and the DC-SIGN

To further understand the biological role of the *G. morsitans* salivary *N*-glycans, we explored their potential recognition by cells from the immune system. Endocytic c-type lectin receptors, such as macrophage mannose receptor (MR—CTLD) and the dendritic cell-specific ICAM3 grabbing nonintegrin (DC-SIGN), can recognize exposed mannose residues on glycoproteins. Using recombinant CTLD4-7-Fc and recombinant Human DC-SIGN Fc Chimera proteins, the carbohydrate-binding domains from these two receptors, we performed overlay assays using saliva before and after treatment with PNGase F (Fig 6). Our results showed that CTLD4-7-Fc recognized at least 4 glycoprotein bands migrating around 45–75 kDa in *G. morsitans* saliva, while DC-SIGN recognized only two of them around 70 kDa. Recognition of these bands by either lectin disappeared after PNGase F treatment, confirming specificity of binding to *N*-linked mannosylated glycans. However, the overall recognition of tsetse salivary glycoproteins by either lectin is much lower compared to that observed with Con A (Fig 4, lane 16).

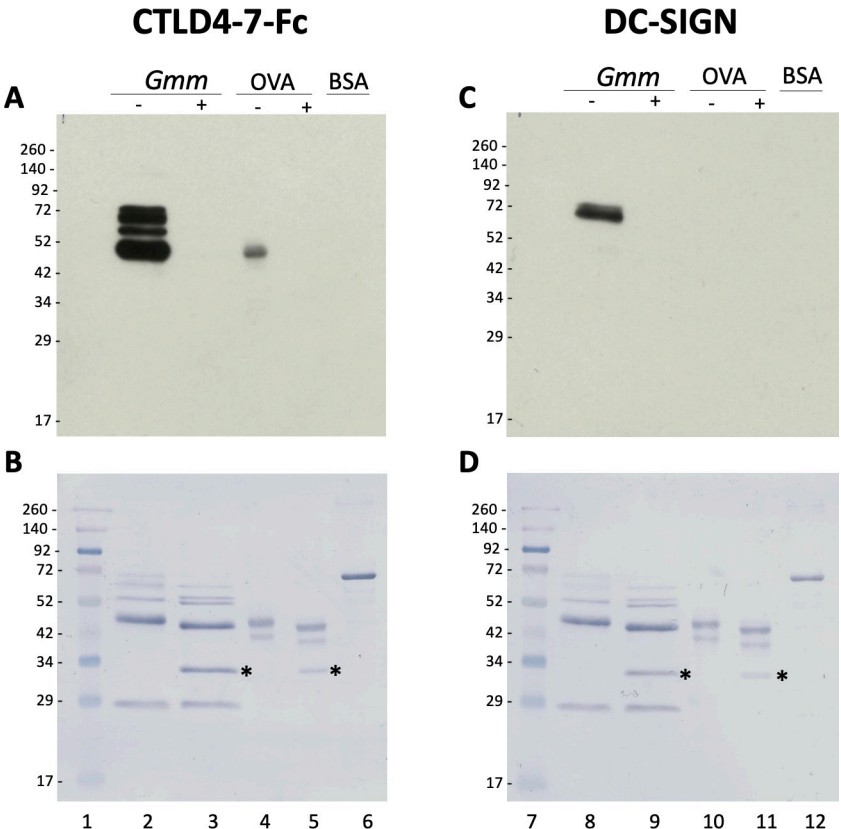

**Fig 6. Tsetse salivary *N*-glycans are recognized by C-type lectins Mannose Receptor and DC-SIGN.** 2 µg of *Glossina morsitans* saliva (*Gmm*) were untreated (-) or treated (+) with PNGase F and then processed for overlay assays using either recombinant CTLD4-7-Fc (A) or DC-SIGN (B). MWM, lanes 1 and 7; *Gmm* saliva, lanes 2, 3, 8 and 9; OVA, egg albumin positive control (lanes 4, 5, 10 and 11); BSA, bovine serum albumin negative control (lanes 6 and 12). Nigrosine-stained membranes (B, D) are shown as loading controls for (A) and (B), respectively. Asterisk indicates PNGase F enzyme.

## Discussion

We revealed for the first time the composition and structure of the oligosaccharides modifying the salivary proteins of the tsetse fly *G. m. morsitans*, vector of African trypanosomiasis. Through enzymatic analysis coupled with highly sensitive LC and ESI-MS/MS, we found that salivary *N*-glycans in *G. morsitans* saliva are mainly represented by the paucimannose $Man_3GlcNAc_2$ structure, with the addition of some high-mannose glycans and a few hybrid-type oligosaccharides. Three hybrid structures with terminal GlcNAc residues were also detected: $Man_3GlcNAc_3$, $Man_4GlcNAc_3$ and $Man_5GlcNAc_3$. In addition, we only found one fucosylated structure ($FucMan_3GlcNAc_2$), indicating that most of the sugars do not undergo further processing after trimming of the mannose residues.

Research on insect glycans shows these are mainly oligomannose, although complex structures with terminal GlcNAc are often found in some species [12]. Core fucosylation (both α1–3 and α1–6) is quite common in invertebrate glycans [33], and can sometimes cause allergic reactions in vertebrates [34]. Overall, tsetse salivary *N*-glycans fit these observations; however, we did not find evidence of α1–3 linked fucosylation in the saliva of this fly. Some studies have suggested *G. m. centralis* saliva contained as many as seven glycoproteins, and had a high proportion of mannose [35], while *G. m. morsitans* contained four salivary glycoprotein bands

[28], and predicted *N*-glycosylation sites in TSGF 1 and 2 [36], and Gmmsgp 2 and 3 [16]. Caljon and colleagues (2010) reported on the putative glycosylation of salivary 5'nucleotidase-related apyrase (5'Nuc), a salivary protein that interrupts formation of the hemostatic plug by hydrolyzing ATP and ADP [37]. The NetNGlyc server identified four glycosylation sites in the peptide sequence, consisting with the ~5 kDa loss in mass after PNGase F treatment (in agreement with our results) [37]. Assays with recombinant non-glycosylated form of 5'Nuc suggested sugars are not essential for its activity, but they might be important for secretion and solubility. However, the role of these glycan modifications in the tsetse salivary proteins remains to be elucidated; further characterization of the *Glossina N*-glycome should include site-specific analyses in order to better understand the influence of glycosylation in the role of the salivary protein.

The presence of glycoproteins has also been described in the salivary glands of *An. gambiae* (some being female-specific) [38], *An. stephensi* [39], *Ae. albopictus* [40], and *Phlebotomus duboscqi* sandflies [41]. However, the most complete structural characterization of salivary glycoproteins in disease vectors to date has been that of *Lutzomyia longipalpis*, vector of visceral leishmaniasis in the Americas [42]. This sand fly species makes mostly oligomannose *N*-glycans, with $Man_5GlcNAc_2$ being the most abundant structure. When we compare the salivary *N*-glycomes of *G. morsitans* with that of the sand fly *Lu. longipalpis*, it is possible to observe that both profiles are strikingly similar regarding the high content of mannosylated species, except that in tsetse flies the major glycan is the tri-antennary core $Man_3GlcNAc_2$ structure, compared to the dominant $Man_5GlcNAc_2$ in sand flies. In general, the dominance of mannosylated *N*-glycans that was found in both species suggests a conserved protein glycosylation pathway among hematophagous dipterans; in addition to potentially modulating pathogen transmission, this raises the question on the functional role(s) these mannosylated sugars may have during insect blood feeding.

The abundance of paucimannose and oligomannose sugars in tsetse salivary glycoproteins leads us to hypothesize about how these might interact with the host immune system. Several cells of the dermal immunological repertoire harbor receptors with carbohydrate binding domains [43], such as the mannose receptor (MR), which is expressed in populations of macrophages and dendritic cells, and participates in antigen presentation and the clearance of molecules [44]. The C-type lectin-like domain (CTLD) of the MR binds glycosylated molecules with terminal Man, Fuc or GlcNAc; our work shows that a recombinant CTLD4 can recognize several *N*-glycans from tsetse saliva. Another example of these receptors is the DC-SIGN, a dendritic cell receptor involved in antigen presentation and the initial detection of pathogens. Its carbohydrate recognition domains bind to high-mannose oligosaccharides, which mediate dendritic cell recognition of pathogens like *Mycobacterium tuberculosis* and *Leishmania* [45, 46]. We also show here that a recombinant fraction of the DC-SIGN recognizes some salivary glycoproteins from tsetse saliva. Notably, the DC-SIGN seems to recognize fewer tsetse glycoproteins than the mannose receptor, possibly due to its specificity for high-mannose glycan structures [47], which would not bind to the most abundant $Man_3GlcNAc$ glycan species.

Tsetse saliva is known to produce different effects in the skin of the vertebrate host [16, 18, 37, 48], which can vary between first-time and chronic exposure [35]; however, we don't know what roles protein-modifying sugars have in these processes. Salivary glycans might also be interacting with the lectin complement system, which is initiated when the mannan-binding lectin or ficolins bind to carbohydrates on pathogens. Mannan-binding lectin binds to several sugars, including Man, GlcNAc and Fuc, through which it recognizes pathogens like *T. cruzi*, *Leishmania* and *Plasmodium* [49]. Ficolins on the other hand bind to glycans containing disulfated *N*-acetyllactosamine, terminal Gal or GlcNAc [50]. Although there are no reports of this pathway activation by either *Glossina* or *Trypanosoma*, the potential masking of metacyclics

with salivary glycoproteins could be a mechanism to reduce destruction by the lectin pathway of complement or activation of DC-SIGN-dependent immune response.

Interestingly, a tsetse salivary gland infection with *T. brucei* did not alter the *N*-glycosylation profile in saliva samples, suggesting that it does not seem to affect the biosynthesis or transfer of *N*-glycans to proteins in salivary epithelial cells. This was a surprising finding given that trypanosomes cause such a profound transcriptional downregulation of most tsetse salivary proteins [16]. Furthermore, when we investigated the immune reactivity of salivary proteins with antibodies using an anti-*G. morsitans* polyclonal serum, a larger number of antigenic epitopes were detected in high molecular weight proteins only after PNGase F deglycosylation (more evident in trypanosome-infected samples). It is possible that some salivary epitopes are being masked from the immune system by glycans. Even though parasite infection does not affect salivary *N*-glycosylation, we noted slight variations in the relative abundance of some glycan structures in unfed versus bloodfed flies; this could be due to the bloodmeal itself causing changes in the salivary protein glycosylation, an effect that would be interesting to explore further.

In this work we also evaluated the binding of salivary proteins to the metacyclic trypanosomes (which infect the vertebrate host). We unexpectedly found that that only glycosylated salivary proteins (e.g. 5'Nucleotidase related protein Gmmsgp3 and TSGF and Tsal) associate with the trypanosome surface. The biological significance of this finding remains to be determined, but it may enable immunomodulatory activity in the immediate parasite microenvironment during the early infection processes in the skin [48]. In some cases, a vector's salivary proteins may associate to the surface of a pathogen, and affect how the vertebrate host's immune system recognizes and eliminates the invader. An example of this is the Salp15 salivary glycoprotein of *Ixodes* ticks, which binds to the outer surface protein C of *Borrelia burgdorferi*, creating a protective coat against complement-mediated killing [51–53]. There also evidence that some *Aedes* salivary proteins may interact with dengue virions to favor their transmission [54, 55]. MosGILT, a protein from the salivary glands of *Anopheles*, also binds to the surface of *Plasmodium* sporozoites; however, in this case it negatively affects the traversal activity of parasites and reduces their ability to infect the host [56].

To evaluate the presence of *O*-linked glycans in tsetse salivary glycoproteins, we released the *O*-glycans by hydrazinolysis [24] and analyzed fluorescently labelled samples by HILIC-HPLC and MS. However, we did not detect *O*-glycans by either HILIC or MS (S6 Fig), which coincides with our recent findings in the saliva of *Lutzomyia longipalpis* sand flies [42]. This suggests that either these sugars are made in very low abundance by salivary gland cells of these insects, or they may be structurally different compared to mammalian *O*-glycans. Nevertheless, the *G. morsitans* sialome describes the presence of at least nine mucin polypeptides, which are members of the hemomucin family [17], and predictions showed that these proteins have anywhere between 12 and 40 putative *O*-linked glycosylation sites. In addition, *Glossina* species express a large family of peritrophins and peritrophin-like glycoproteins in the peritrophic matrix [57, 58], which protects the fly from harmful components present in the bloodmeal. Tsetse peritrophins contain one or more mucin domains that are likely modified with *O*-linked glycosaminoglycans (GAGs) [59]. It remains to be seen whether insect salivary glycoproteins, containing mucin domains, are modified by GAGs or completely lack canonical (GalNAc-linked) mammalian *O*-glycosylation.

## Supporting information

**S1 Fig. Schiff's staining analysis of *G. morsitans* salivary glycoproteins.** 10 μg *G. morsitans* salivary proteins (lanes 1 and 2) and egg albumin (lanes 3 and 4) were incubated overnight

with (+) or without (-) PNGase F to cleave N-glycans. Samples were resolved on a 12% SDS-PAGE gel and stained with Colloidal Coomassie Blue (A) or Schiff's (B) staining. Asterisk indicates PNGase F enzyme.
(DOCX)

**S2 Fig. Endo H cleavage of *G. morsitans* salivary glycoproteins.** 10 μg *G. morsitans* salivary proteins (lanes 1 and 2) and egg albumin (lanes 3 and 4) were incubated overnight with (+) or without (-) Endo H. Samples were resolved on a 12% SDS-PAGE gel and Coomassie stained. There was a notable shift in migration in 4 bands (1–4) after deglycosylation. These bands were excised, trypsinised and identified by mass spectrometry. 1, 5' Nucleotidase; 2, TSGF 2/ Adenosine deaminase; 3, TSGF 1; 4, Tsal 1/2. Asterisk indicates Endo H.
(DOCX)

**S3 Fig. HILIC-UHPLC chromatograms of 2-AB labelled *N*-glycans released enzymatically from teneral fly saliva.** (A) *N*-glycans released by PNGase A, (B) *N*-glycans released by PNGase F, (C) *N*-glycans released by PNGase A after deglycosylation with PNGase F. Green circle, mannose; blue square, *N*-acetylglucosamine; red triangle, fucose. Peaks labelled with an asterisk refer to buffer contaminants.
(DOCX)

**S4 Fig. Positive-ion ESI-MS/MS fragmentation spectra of procainamide-labelled *N*-glycans from teneral tsetse fly saliva.** Spectra correspond to (A) *m/z* 1130.49 (Man$_3$GlcNAc$_2$-Proc), (B) *m/z* 1276.52 (Man$_3$GlcNAc$_2$Fuc-Proc), (C) *m/z* 1333.57 (Man$_3$GlcNAc$_3$-Proc). Green circle, mannose; blue square, *N*-Acetylglucosamine; red triangle, fucose; Proc, procainamide.
(DOCX)

**S5 Fig. SDS-PAGE fractionation of 10 μg of *G. morsitans* salivary profiles obtained from different infection stages.** Lane 1, teneral; Lane 2, 4-week old, bloodfed flies (Bloodfed); Lane 3, flies with salivary gland *T. brucei* infection (Salivary inf.); Lane 4, flies with midgut *T. brucei* infection (Midgut inf.).
(DOCX)

**S6 Fig. HILIC-UHPLC profiles of 2-AB labelled material released by hydrazinolysis.** (A) Glucose homopolymer (GHP); (B) *O*-glycans released from bovine fetuin positive control; (C) material released from teneral tsetse saliva. The peaks labelled with asterisks (*) correspond to hydrazinolysis side products (also present in the fetuin and water control samples). (D) water blank (negative control). GU, glucose homopolymer ladder. 2-AB, 2-aminobenzamide.
(DOCX)

**S1 Table. Prediction of potential *N*-linked glycosylation sites in *Glossina morsitans morsitans* salivary proteins.** Protein sequences were retrieved from Vector Base (https://vectorbase. org) and then searched on the NetNGlyc 1.0 server (http://www.cbs.dtu.dk/services/ NetNGlyc/) to find N-X-S/T sequons. Signal Peptide was predicted using the SignalP-5.0 server (http://www.cbs.dtu.dk/services/SignalP/).
(XLSX)

**S2 Table. Proteomic identification of tsetse salivary glycoproteins susceptible to PNGase F cleavage (Fig 1).**
(DOCX)

**S3 Table. Proteomic identification of tsetse salivary proteins susceptible to Endo-H treatment.**
(DOCX)

**S4 Table. HILIC- LC-ESI-MS data with sugar composition and structures for *N*-glycans released by PNGase F and labelled with procainamide.** The symbols for glycan structures are adopted from the Consortium for Functional Glycomics (http://www.functionalglycomics.org/). Red triangle, Fuc; blue square, GlcNAc; green circle, Man; Hex, hexose; HexNAc, *N*-acetylhexosamine.
(DOCX)

**S5 Table. HILIC-LC-ESI-MS/MS data for *N*-glycans released by PNGase F.** Table shows details for three representative glycan structures from saliva from teneral flies. *Mass corresponds to loss of diethylamine ion (73 Da).
(DOCX)

# Acknowledgments

We thank Prof. Wendy Gibson for supplying TSW-196 trypanosome stabilates, Dr. Luisa Martínez-Pomares for the recombinant CTLD-4-7-Fc, and Mr. Douglas Lamont (Dundee University Fingerprints Proteomics Facility) for protein identification.

# Author Contributions

**Conceptualization:** Radoslaw P. Kozak, Karina Mondragon-Shem, Christopher Williams, Guy Caljon, Jan Van Den Abbeele, Álvaro Acosta-Serrano.

**Data curation:** Radoslaw P. Kozak, Karina Mondragon-Shem, Christopher Williams, Clair Rose, Samirah Perally, Guy Caljon, Jan Van Den Abbeele, Katherine Wongtrakul-Kish, Richard A. Gardner.

**Formal analysis:** Radoslaw P. Kozak, Karina Mondragon-Shem, Christopher Williams, Clair Rose, Samirah Perally, Guy Caljon, Jan Van Den Abbeele, Katherine Wongtrakul-Kish, Richard A. Gardner, Daniel Spencer, Michael J. Lehane, Álvaro Acosta-Serrano.

**Funding acquisition:** Daniel Spencer, Álvaro Acosta-Serrano.

**Investigation:** Radoslaw P. Kozak, Karina Mondragon-Shem, Christopher Williams, Clair Rose, Samirah Perally, Guy Caljon, Jan Van Den Abbeele, Katherine Wongtrakul-Kish, Richard A. Gardner, Daniel Spencer, Álvaro Acosta-Serrano.

**Project administration:** Álvaro Acosta-Serrano.

**Resources:** Guy Caljon, Jan Van Den Abbeele, Daniel Spencer, Álvaro Acosta-Serrano.

**Supervision:** Daniel Spencer, Álvaro Acosta-Serrano.

**Writing – original draft:** Radoslaw P. Kozak, Karina Mondragon-Shem, Christopher Williams, Álvaro Acosta-Serrano.

**Writing – review & editing:** Radoslaw P. Kozak, Karina Mondragon-Shem, Álvaro Acosta-Serrano.

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
