## [Decision Letter · Decision Letter 0]

3 Nov 2020

Dear Dr. Acosta-Serrano,

Thank you very much for submitting your manuscript "Tsetse salivary glycoproteins are modified with paucimannosidic N-glycans, are recognised by C-type lectins and bind to trypanosomes" for consideration at PLOS Neglected Tropical Diseases. As with all papers reviewed by the journal, your manuscript was reviewed by members of the editorial board and by several independent reviewers. The reviewers appreciated the attention to an important topic. Based on the reviews, we are likely to accept this manuscript for publication, providing that you modify the manuscript according to the review recommendations. 

The manuscript is a well-performed and well-presented study that contains novel and useful data. It has received favorable reviews from three expert Referees. The authors are asked to make a number of changes, suggested by Reviewer 2 mainly. These changes will improve the overall presentation and readability of the work, in particular for non-specialists. Among these changes, the authors should pay particular attention to the O-glycan aspect: (i) they should be careful not to induce the reader think that the N-glycome analyzed is equivalent to the overall glycome of the tse tse fly saliva, (ii) if possible, they should detail the (negative) results they obtained for O-glycans and (iii) even if they do not present the negative O-glycan data, in the Discussion they should make it explicit how O-glycan release was attempted (as otherwise the discussion on possible structural causes for the lack of detection of O-glycans is difficult to follow).

Sincerely,

Alvaro Diaz, PhD

Associate Editor

Paula MacGregor

Deputy Editor

The manuscript is a well-performed and well-presented study that contains novel and useful data. It has received favorable reviews from three expert Referees. The authors are asked to make a number of changes, suggested by Reviewer 2 mainly. These changes will improve the overall presentation and readability of the work, in particular for non-specialists. Among these changes, the authors should pay particular attention to the O-glycan aspect: (i) they should be careful not to induce the reader think that the N-glycome analyzed is equivalent to the overall glycome of the tse tse fly saliva, (ii) if possible, they should detail the (negative) results they obtained for O-glycans and (iii) even if they do not present the negative O-glycan data, in the Discussion they should make it explicit how O-glycan release was attempted (as otherwise the discussion on possible structural causes for the lack of detection of O-glycans is difficult to follow).

Reviewer's Responses to Questions

**Key Review Criteria Required for Acceptance?**

**Methods**

-Are the objectives of the study clearly articulated with a clear testable hypothesis stated?

-Is the study design appropriate to address the stated objectives?

-Is the population clearly described and appropriate for the hypothesis being tested?

-Is the sample size sufficient to ensure adequate power to address the hypothesis being tested?

-Were correct statistical analysis used to support conclusions?

-Are there concerns about ethical or regulatory requirements being met?

Reviewer #1: Work presents a good balance of biochemical and spectrometric techniques.

-Are the objectives of the study clearly articulated with a clear testable hypothesis stated? Yes.

-Is the study design appropriate to address the stated objectives? Yes.

-Is the population clearly described and appropriate for the hypothesis being tested? Yes.

-Is the sample size sufficient to ensure adequate power to address the hypothesis being tested? Yes.

-Were correct statistical analysis used to support conclusions? Yes.

-Are there concerns about ethical or regulatory requirements being met? NA.

Reviewer #2: In this manuscript the authors study the structures of the N-glycans present in salivary glycoproteins from tsetse flies before and after infection with trypanosome and their binding to the parasite surface. The analyses conducted were appropriate and conclusive. 

Biological experiments met ethical and regulatory requirements.

Reviewer #3: In this manuscript, Kozak et al. present the structural characterization of the salivary N-glycome from Glossina morsitans (vector of African Trypanosomiasis), adding a functional perspective on how it may influence parasite transmission into the vertebrate host. In order to do so, the authors characterize the N-glycosylation profile in Tse Tse fly salivary N-glycoproteins. The authors provide a detailed analysis on the salivary N-glycome via state-of-the-art methodologies. Going forward, they study the influence of T. brucei infection on the salivary N-glycan profile, and to give an insight into its functional role and its interactions with the vector and the host immune system, they study potential interactions of these N-glycosylated proteins with T. brucei and also with lectins relevant to the immune response, such as mammalian C-type lectins (in particular, mannose receptor and DC-SIGN). 

Considering the scarce information on the subject, and being the first structural characterization of the salivary N-glycome in this insect, I think the manuscript is interesting and could be accepted after careful revision. The study design is appropriate and there are no ethical issues.

**Results**

-Does the analysis presented match the analysis plan?

-Are the results clearly and completely presented?

-Are the figures (Tables, Images) of sufficient quality for clarity?

Reviewer #1: -Does the analysis presented match the analysis plan? Yes.

-Are the results clearly and completely presented? Yes.

-Are the figures (Tables, Images) of sufficient quality for clarity? Yes.

Reviewer #2: The authors performed a battery of studies, including mass spectrometry, UHPLC, hydrophilic LC-MS and treatment with exoglycosidases and lectins to achieve the structural determination of the N-glycans present in salivary proteins. The results are thoroughly presented in the main manuscript and the supplementary files. 

Biological experiments represent an initial approach to the understanding of the interaction between the parasite and the host immune system and CTLD AND DC-SIGN binding assays are consistent to the glycan structures found.

Reviewer #3: All figures and tables are clear and in good quality, most of the details observed can be ascribed to the .pdf format (i.e. detailed structure of N-glycans described in Table 1 are outside cell limits or moved). 

Please see next section for suggestions on how to improve presentation of data. 

Given the originality of the results pertaining the characterization of the N-glycome in the saliva of Glossina morsitans (all previous work on insect glycosylation has been mainly done in the model fly Drosophila melanogaster, and particularly considering G. morsitans, previous work was solely based on predictions by sequence or partial characterization by lectin blots, with no structural analysis), some of the supplementary data should be added as an additional figure to the main manuscript. A detailed description on specific points to be addressed is included in following sections.

**Conclusions**

-Are the conclusions supported by the data presented?

-Are the limitations of analysis clearly described?

-Do the authors discuss how these data can be helpful to advance our understanding of the topic under study?

-Is public health relevance addressed?

Reviewer #1: -Are the conclusions supported by the data presented? Yes.

-Are the limitations of analysis clearly described? Yes.

-Do the authors discuss how these data can be helpful to advance our understanding of the topic under study? Yes.

-Is public health relevance addressed? Yes.

Reviewer #2: The results from the structural determinations are conclusive and, although no significant differences between N-glycosylation of infected and non-infective salivary proteins were found, the knowledge of this fact contribute to the understanding of host-parasite interaction. Biological tests are more preliminary and encourage further studies on the subject. Particularly, since glycans seem to be essential to the binding of the protein to the parasite it would be interesting to determine which structures are present in the most abundant bound protein.

Reviewer #3: The conclusions are supported by the data, although limitations of the N-glycosylation analysis performed should be stated in the discussion. The data presented is very valuable, but it is not a site-specific analysis, which could be a second step in the future for a thorough characterization of the N-glycoproteins described here, and could help understand the differences in the results pertaining lectin recognition. 

Moreover, and considering that data on O-glycosylation is not shown, all the manuscript should be thoroughly revised and corrected in order not to induce the reader on the misconcept that the salivary glycome of the Tse Tse fly is characterized. The text reads “glycome, glycans, glycosylation”, but the study is limited to the N-glycome, N-glycans, and N-glycosylation. Even though the authors do mention that they perform an O-glycan analysis by HILIC or MS with negative results, the data is not shown and only mentioned in the Discussion.

**Editorial and Data Presentation Modifications?**

Reviewer #1: Please cite figure 3B in the text 

Figure 4, indicate the salivary proteins in the metacyclic trypanosome lysate with an asterisks as stated in the figure legend.

Reviewer #2: In the conclusions the authors stated that they did not found O-glycosylation. Even though it is a no shown result, it should be mention how this data was obtained. That is, if the glycoproteins were treated chemically or enzymatically to release the O-linked glycans.

Reviewer #3: a) General: 

All text should be corrected in order not to induce the reader on the misconcept that the salivary glycome of the Tse Tse fly is characterized. The text reads “glycome, glycans, glycosylation” in many sentences and sections, and the study is limited to the N-glycome, N-glycans, and N-glycosylation. Even though the authors do mention that they did not detect O-glycans, the data is not shown and only mentioned in the Discussion (Line 543). I believe that the manuscript should be more specific on the specific data on N-glycosylation provided as results.

b) Materials and Methods

Line 249: Streptococcus pneumonia should read Streptococcus pneumoniae

Line 250: Jack Bean Mannosidase is abbreviated as JBM in all the text, apart from this section, where it is defined as JBAM. For the sake of clarity, I suggest homogenizing this with the rest of the text.

In this section, and considering that bkF and JBM are not compatible in their assay requirements, a brief description on the sequential digestion bkF+JBM should be added. 

Line 255: title should read HILIC-UHPLC-FLR, as N-glycans are specifically detected by tagging with a fluorescent probe.

Results: 

Section: Identification of tsetse salivary glycoproteins

I consider that Supplementary Figures 1-3 associated to this section should be incorporated in the main manuscript. Experimental data on N-glycosylated proteins in saliva of Glossina morsitans is rather scarce and is mostly based on predictions, so I think this part of the paper should be incorporated as a Figure. 

Line 317: “Taken together, these results indicate the presence of few key glycoproteins in tsetse fly saliva, and more importantly, the presence of several high mannose or hybrid type N-glycans.” Authors should consider the possibility of lower abundance N-glycosylated proteins that were not detected here as by technical limitations. The final statement should be restricted to those glycoproteins characterized. 

Line 323: As it is referring to 2AB-labelled N-glycans, it should read HILIC-UHPLC-FLR. Also in the legend Figure 1, reads HILIC-(U)HPLC, please homogenize abbreviations.

Figure 1: In the two panels the authors use an asterisk to mark different issues. In Panel A, they highlight buffer contaminants. In Panel B, they mark the m/z 1130.55 as [M+2H]2+ ion, which corresponds to the most abundant N-glycan (Man3GlcNAc2), which has two ions associated to it in this panel. I would advice to use different ways to highlight these issues, as they are very different and it can be confusing to the reader. 

Considering that the peak at around 5 minutes RT in Figure 1A appears with different intensities in other chromatograms, it should be marked in all of them.

Moreover, and regarding the structure Man3GlcNAc2, I believe a short sentence explaining why this particular structure appears as a double peak could help the MS non-specialist reader understand the reasons behind this. In the Results describing Figure 1B this peak is not discussed and only appears in the description on the fragmentation data (Supplementary Figure S5).

A brief explanation for the CFG glycan symbology is required in the legend for this figure, it is only presented in Supplementary Table 4, and without a reference, which should be added for non-specialized readers. 

The chromatograms are shown with retention time as an X axis, and Table 1 describe GU values. There is no description or reference in Materials and Methods to the usage of Dextran as external standard or how GUs are calculated. GUs are not described in the Figure. This can be thoroughly confusing for a non-specialized reader, considering the broad scope of this Journal.

Peak 13 (Man9GlcNAc2) described in 1A is missing in 1B, no explanation for this.

Table 1: Exoglycosidase digestion of 2-AB-labelled glycans released from teneral fly saliva by PNGase F.

The first column should read HILIC-UHPLC, as retention times can be very different in HPLC.

I think it would really improve the presentation of data if authors could add a new column with the MS data from Figure 1B, so the reader can integrate the information. I understand MS analysis is based on procainamide-labelled N-glycans, but this can be specified in the Table. In Figure 1A, and to complement this table, I would indicate with arrows the shifts in RT caused by digestions. Finally, I would also add a column before the structure column with the abbreviation used in the text for each glycan (the text uses abbreviations such as Man3GlcNAc2 with few references to the peak numbers, and in the table only the CFG symbology and peak number is used).

Line 381 and following paragraph: “We then determined whether T. b. brucei infection alters the structure of salivary N-glycans. Oligosaccharides were released with PNGase F, labelled with procainamide, and then analyzed by HILIC coupled with ESI-MS. Figure 2A shows the HILIC chromatogram, where no change is observed in the saliva N-glycan population following infection of the salivary gland. A comparison of the relative percentage areas of glycan peaks identified in teneral, naïve and infected fly saliva (Table 2) showed no quantitative differences after infection either.” 

In Figure 2, chromatograms look exactly the same as those in Figure 1, and corresponding legend states that the chromatograms are from 2AB-labelled glycans. The text should be corrected, as from this paragraph, it seems that chromatograms are based on procrainamide labelling, not 2AB. 

Peak 13 is again missing in 2B. If it cannot be seen in the MS due to technical issues (i.e. low ionization), please indicate so, as it is also missing in Figure 1B.

The description of the “no differences” in N-glycosylation following infection is quite confusing. 

“A comparison of the relative percentage areas of glycan peaks identified in teneral, naïve and infected fly saliva (Table 2) showed no quantitative differences after infection either.”

There are differences between teneral and naïve N-glycans, there are no differences between infected and naïve samples. Analysis should be rephrased by pairing of comparable samples (teneral vs naïve, naïve vs bloodfed).

“This is confirmed by the MS spectra (Figure 2B) which shows that salivary glycan structures and abundances remain unaffected by infection with T. brucei. However, when comparing teneral versus naïve (bloodfed) flies or infected flies, there seemed to be a slight variation in the abundance of some structures (Table 2: peaks 4, 9, 10, 13), potentially an effect of blood ingestion.”

Just to follow up on the previous comments, peaks in this sentence are only defined by number, and not by name. Moreover, how can differences/similarities in abundance be quantified from Figure 2B? Have the authors compared intensities for procrainamide labelled N-glycans obtained from infected and naïve samples using the BPI data channel? That is not shown in Figure 2B. 

Table 2: Comparison of relative abundance of released N-glycans present in saliva samples from teneral (unfed), naïve and T. brucei-infected flies. 

As per materials and methods, it should read HILIC-UHPLC instead of HPLC. Here again GU values are used and not described. Moreover, it should clarify that it was done with 2AB.

Line 385: Teneral and naïve should be defined as this is the first part of the results where they are used. They are generally defined in Materials and Methods, Figure Legends and Tables, but it would be really useful to have it defined in the Results as first time mentioned. 

Discussion: 

Line 457: "Through enzymatic analysis coupled with highly sensitive LC and ESI-MS/MS, we found that salivary N-glycans in G. morsitans saliva are mainly represented by the paucimannose Man3GlcNAc2 structure, with the addition of a few hybrid type oligosaccharides."

This sentence should be corrected as high mannose N-glycans are also detected.

Line 468: When comparing to data on other insects, absence of a(1,3) fucosylated N-glycans in G. morsitans (shown by the authors by digestion with PNGase A) should be present in the discussion.

Line 470: are sgp3 and Gmmsgp3 the same? If so, please homogenize the abbreviation.

Line 502 and corresponding paragraph: “We also show here that a recombinant fraction of the DC-SIGN recognizes some salivary glycoproteins from tsetse saliva. Notably, the DC-SIGN seems to recognize fewer tsetse glycoproteins than the mannose receptor, possibly due to its specificity for high-mannose glycan structures [43], which would not recognize the most abundant Man3GlcNAc glycan species.”

Even though I understand this is a first approach to interactions with soluble forms of MR and DC-SIGN, a correlation with the N-glycoproteins identified in the previous sections of the manuscript could be provided, in order to integrate this data with all previous work in the manuscript. 

Data published in the following paper could also be relevant to the discussion of the results:

Ellis JA, Shapiro SZ, ole Moi-Yoi O, Moloo SK. Lesions and saliva-specific antibody responses in rabbits with immediate and delayed hypersensitivity reactions to the bites of Glossina morsitans centralis. Vet Pathol. 1986 Nov;23(6):661-7. doi: 10.1177/030098588602300603. PMID: 3811131.

Finally, and as previously stated, the discussion should consider the limitations of de-N-glycosylation in identifying specific structures for each glycoprotein and N-glycosylation site, something that can be achieved by site-specific analysis. 

Supplementary data:

Title in supplementary data does not correspond to the title in the Manuscript. 

Supplementary Tables 1 and 2: It would be easier to follow the results if authors could indicate the same band numbers in Figures and Tables. 

The reference “Li and Aksoy, 2000” is not cited in the References. Is this one?

Song Li, Serap Aksoy, A family of genes with growth factor and adenosine deaminase similarity are preferentially expressed in the salivary glands of Glossina m. morsitans, Gene, Volume 252, Issues 1–2, 2000, Pages 83-93, ISSN 0378-1119, https://doi.org/10.1016/S0378-1119(00)00226-2

The reference “Li et al., 2001” is not cited in references. Is this one?

Li S, Kwon J, Aksoy S. Characterization of genes expressed in the salivary glands of the tsetse fly, Glossina morsitans morsitans. Insect Mol Biol. 2001 Feb;10(1):69-76. doi: 10.1046/j.1365-2583.2001.00240.x. PMID: 11240638.

Apart from these two missing references, Ref. 1 and 3 in Supplementary Data are Ref. 33 and 17, respectively, in the main text. 

All references in Supplementary Data should be included and correlatively numbered with the main reference list. If not present there, and if the Journal allows it, they could also be added a list of supplementary references.

Figure S4: For better comparison, structures or GU values should be added. Peak around 5 min should be marked as a contaminant. Legend reads: "Figure S4. HILIC-(U)HPLC chromatograms of 2-AB labelled N-glycans released enzymatically from teneral fly saliva. (A) N-glycans released by PNGase A, (B) N-glycans released by PNGase F, (C) Nglycans released by PNGase A after deglycosylation with PNGase F. The glycan structures corresponding to the numbers on the HPLC peaks are listed in Table 3."

There is no Table 3 (corresponds to Table 1?), and the chromatography corresponds to HILIC-UPLC-FLR, not HPLC.

**Summary and General Comments**

Reviewer #1: In this study, the authors have presented the first structural analysis of salivary glycans from tsetse flies that have economic and medical impact in sub-Saharan Africa as the biological vectors of trypanosomes, which cause human sleeping sickness and animal trypanosomiasis. It is a nice, well done, work of glycocobiology. 

Using enzymatic analysis

coupled with highly sensitive LC and ESI-MS/MS authors show that salivary glycoproteins are mainly modified by mainly of pauci-mannose and high-mannose N-glycans, which are recognized by mammalian C-type lectins. Furthermore, they show that salivary glycoproteins bind to the surface of the trypanosomes that are transmitted to the vertebrate host.

Reviewer #2: The manuscript contains a solid research work that contributes to the knowledge of the tsetse glycome and a preliminary approach to determine the interaction with the parasite and the host immune system.

Reviewer #3: (No Response)

PLOS authors have the option to publish the peer review history of their article (what does this mean?). If published, this will include your full peer review and any attached files.

Reviewer #1: Yes: Adriane R. Todeschini

Reviewer #2: No

Reviewer #3: No
---

## [Editor Report · Decision Letter 1]

14 Dec 2020

Dear Dr. Acosta-Serrano,

We are pleased to inform you that your manuscript 'Tsetse salivary glycoproteins are modified with paucimannosidic N-glycans, are recognised by C-type lectins and bind to trypanosomes' has been provisionally accepted for publication in PLOS Neglected Tropical Diseases.

Best regards,

Alvaro Diaz, PhD

Associate Editor

Paula MacGregor

Deputy Editor

---

## [Editor Report · Acceptance letter]

25 Jan 2021

Dear Dr. Acosta-Serrano,

We are delighted to inform you that your manuscript, "Tsetse salivary glycoproteins are modified with paucimannosidic * N*-glycans, are recognised by C-type lectins and bind to trypanosomes," has been formally accepted for publication in PLOS Neglected Tropical Diseases.

Best regards,

Shaden Kamhawi

co-Editor-in-Chief

Paul Brindley

co-Editor-in-Chief
